# RSK1 vs. RSK2 Inhibitory Activity of the Marine β-Carboline Alkaloid Manzamine A: A Biochemical, Cervical Cancer Protein Expression, and Computational Study

**DOI:** 10.3390/md19090506

**Published:** 2021-09-07

**Authors:** Alejandro M. S. Mayer, Mary L. Hall, Joseph Lach, Jonathan Clifford, Kevin Chandrasena, Caitlin Canton, Maria Kontoyianni, Yeun-Mun Choo, Dev Karan, Mark T. Hamann

**Affiliations:** 1Department of Pharmacology, College of Graduate Studies, Midwestern University, 555 31st Street, Downers Grove, IL 60515, USA; mhall1@midwestern.edu (M.L.H.); joseph.m.lach@gmail.com (J.L.); clifford07@gmail.com (J.C.); kschandras@gmail.com (K.C.); canto030@umn.edu (C.C.); 2Department of Pharmaceutical Sciences, Southern Illinois University Edwardsville, Edwardsville, IL 62026, USA; mkontoy@siue.edu; 3Department of Chemistry, University of Malaya, Kuala Lumpur 50603, Malaysia; ymchoo@um.edu.my; 4Department of Pathology, Medical College of Wisconsin, Milwaukee, WI 53226, USA; dkaran@mcw.edu; 5Department of Drug Discovery and Biomedical Sciences and Public Health, Colleges of Pharmacy and Medicine, Medical University of South Carolina, Charleston, SC 29425, USA; hamannm@musc.edu

**Keywords:** MZA, Manzamine A, CTKD, C-terminal kinase domain, ras-ERK1/2 (extracellular-signal-regulated kinase 1/2) pathway, RSK1, 90 kDa ribosomal protein S6 kinase 1, RSK2, 90 kDa ribosomal protein S6 kinase 2, NTKD, N-terminal kinase domain

## Abstract

Manzamines are complex polycyclic marine-derived β-carboline alkaloids with reported anticancer, immunostimulatory, anti-inflammatory, antibacterial, antiviral, antimalarial, neuritogenic, hyperlipidemia, and atherosclerosis suppression bioactivities, putatively associated with inhibition of glycogen synthase kinase-3, cyclin-dependent kinase 5, SIX1, and vacuolar ATPases. We hypothesized that additional, yet undiscovered molecular targets might be associated with Manzamine A’s (MZA) reported pharmacological properties. We report here, for the first time, that MZA selectively inhibited a 90 kDa ribosomal protein kinase S6 (RSK1) when screened against a panel of 30 protein kinases, while in vitro RSK kinase assays demonstrated a 10-fold selectivity in the potency of MZA against RSK1 versus RSK2. The effect of MZA on inhibiting cellular RSK1 and RSK2 protein expression was validated in SiHa and CaSki human cervical carcinoma cell lines. MZA’s differential binding and selectivity toward the two isoforms was also supported by computational docking experiments. Specifically, the RSK1-MZA (N- and C-termini) complexes appear to have stronger interactions and preferable energetics contrary to the RSK2–MZA ones. In addition, our computational strategy suggests that MZA binds to the N-terminal kinase domain of RSK1 rather than the C-terminal domain. RSK is a vertebrate family of cytosolic serine-threonine kinases that act downstream of the ras-ERK1/2 (extracellular-signal-regulated kinase 1/2) pathway, which phosphorylates substrates shown to regulate several cellular processes, including growth, survival, and proliferation. Consequently, our findings have led us to hypothesize that MZA and the currently known manzamine-type alkaloids isolated from several sponge genera may have novel pharmacological properties with unique molecular targets, and MZA provides a new tool for chemical-biology studies involving RSK1.

## 1. Introduction

Marine biological and chemical diversity continue to demonstrate significant potential to contribute novel pharmacology for multiple therapeutic categories [1]. Although the current marine pharmaceutical clinical pipeline consists mainly of compounds developed for cancer chemotherapy [2], several marine natural products have been shown to target a range of protein kinases as well [3]. 

First reported in the Okinawan sponge genus *Haliclona* [4], the manzamine class consists of complex polycyclic marine-derived alkaloids, which possess a fused and bridged tetra- or pentacyclic ring system attached to a β-carboline moiety, and it includes more than 80 additional manzamine-type alkaloids isolated from several sponge genera [5,6]. Manzamines exhibit a range of bioactivities: anticancer [7], immunostimulatory [8], anti-inflammatory [9,10], antiviral [11], antimalarial [12], neuritogenic [13], hyperlipidemia and atherosclerosis suppression in vivo [14,15]. Further, they have been reported to target glycogen synthase kinase-3 (GSK-3), cyclin-dependent kinase 5 (CDK5) [16], and vacuolar ATPases [17]. The unique ability to control proliferation in SIX1-overexpressing cells has expanded the utility of MZA for certain types of cancer cells and may find clinical utility [18]. SIX1 expressing is emerging as an important predictor of poor prognosis and more aggressive forms of cancer [19]. We hypothesized that additional, yet undiscovered molecular targets might be associated with MZA (Figure 1).

The 90 kDa ribosomal S6 kinase, or RSK, is a vertebrate family of cytosolic serine-threonine kinases that contains four homologous isoforms, namely RSK1-4, which act downstream of the ras-ERK1/2 (extracellular-signal-regulated kinase 1/2) pathway [20]. RSK1 and RSK2 adult and embryonic tissue expression has been investigated, with RSK1 found in lung, kidney, pancreas, and brain (cerebellum and microglia) [20,21,22], whereas RSK2 is more abundant in skeletal muscle, heart, pancreas, and brain (neocortex, hippocampus, and cerebellum) [23]. RSK kinases are composed of two functional kinase catalytic domains: the C-terminal kinase domain (CTKD), which belongs to the calcium- and calmodulin-regulated kinases CamK family, is phosphorylated by ERK1/2, and activates the N-terminal kinase domain (NTKD), which belongs to the protein kinase A, G, and C (AGC) family [20]. NTKD has been shown to phosphorylate several substrates, and in turn regulates several cellular processes, including growth, survival, and proliferation [20]. Interestingly, loss of RSK2 function causes a rare form of mental retardation known as Coffin-Lowry syndrome [23], while sustained activation of RSKs appears to be linked to cancer [24,25]. RSK inhibitors used in preclinical studies include two NTKD-interacting inhibitors at the ATP-binding site, namely the reversible dihydropteridinone BI-D1870 [26] and the kaempferol glycoside SL0101 [27], and the pyrrolopyrimidine FMK, a CTKD-interacting irreversible inhibitor that binds to the ATP-binding pocket of RSK [28]. 

Herein we report that screening against a panel of protein kinases revealed MZA inhibited the RSK1 and RSK2 with a high potency toward RSK1. In vitro kinase assays with MZA demonstrated a 10-fold selectivity in potency between RSK1 and RSK2 (relative IC_50_ values of 15.01 and 108.4 μM, respectively). Our biochemical observations were supported by in vitro inhibition of RSK1 and RSK2 by MZA, using two human cervical carcinoma cell lines. Moreover, computational experiments predicted that MZA binds to the ATP-binding pocket of NTKD RSK1, while putative binding affinities of MZA and ATP to the ATP-binding sites of a protein kinase panel offer comparative results pertaining to MZA’s target selectivity. 

## 2. Results 

### 2.1. Effect of MZA on a Protein Kinases Screening Array and Docking Simulations

MZA was screened by using a panel of 30 protein kinases as described [29,30], and the results are shown in Table 1. 

It can be seen that MZA (1 μM) reduced the activity of rat p90 ribosomal S6 kinase 1 or RSK1 by 68%. Furthermore, MZA showed no significant inhibitory effect on the other 29 kinases which were part of the University of Dundee protein kinase enzyme panel. 

Docking experiments were performed in order to predict binding affinities of MZA and ATP for the same panel of kinases in Table 1. Toward that end, we first needed to identify representative crystal structures for RSK-1 and RSK-2, among those deposited in the Protein Databank (pdb). Structural comparisons of respective NTKDs and CTKDs of crystal RSK1 and RSK2 structures (see Methods) led us to select the PDB ID’s in Table 2 for all subsequent docking experiments. It should be noted that we favored the active conformation of NTKD RSK2 (PDB ID: 3G51) [31], as it would be more informative regarding the conformational transitions of the domain. 

Subsequently, docking MZA at the competitive ATP-binding site of the protein kinases was compared against docking of ATP [32]. The predicted binding affinity between MZA and RSK1 is in agreement with the in vitro results (Table 1).

### 2.2. Effect of MZA on RSK1 and RSK2 Protein Kinase Assays 

As depicted in Figure 2, MZA showed moderate yet selective concentration-dependent inhibition of RSK1 (relative IC_50_ = 15.01 ± 1.94 μM) vs. RSK2 (relative IC_50_ = 108.40 ± 16.93 μM) in three independent experiments. In contrast, and as previously reported, BI-D1870, which we used as a positive control, potently inhibited RSK1 and RSK2 (relative IC_50_ = 0.016 ± 0.007 and 0.008 ± 0.009 μM, *n* = 3, respectively), but showed no selectivity toward either RSK1 or RSK2 [24]. 

### 2.3. Effect of MZA on RSK1 and RSK2 in Human Cervical Carcinoma Cell Lines

In support of our in vitro kinase assays and docking experiments (see below), we further examined the biological effect of MZA on RSK1 and RSK2 protein expression levels, using SiHa and CaSki human cervical carcinoma cancer cell lines. As shown in Figure 3, treatment with MZA at 4 μM for 48 h significantly inhibited the protein levels of RSK1 (*p =* 0.002) and RSK2 (*p =* 0.017) compared to DMSO control in the SiHa and CaSki cell line, respectively. Similarly, the use of a higher concentration of MZA (4 and 6 µM) for 48 h completely inhibited phospho-RSK1 protein level. While the anti-phospho-RSK1 antibody recognizes phosphorylated Ser 380 RSK1 protein, the phospho-RSK2 was not detected. These observations support the in vitro protein kinase assays’ results shown in Figure 2 and suggest that MZA may target cellular RSK proteins in human cervical carcinoma cell lines.

### 2.4. Structural Comparisons 

The active sites of NTKD and CTKD RSK1 (PDB ID: 2Z7Q and 3RNY, respectively) with amino acids reportedly involved in binding interactions are depicted in Figure 4a,b. A comparison of the binding pockets in either terminus of RSK1 and RSK2 was performed by overlaying the selected CTKD of RSK1 (PDB ID: 3RNY) onto the CTKD of RSK2 (PDB ID: 4D9T). Figure 4a depicts the equivalent amino acids in the ATP-binding pockets of both structures. Similarly, the overlay of the NTKDs of RSK1 and RSK2 is presented in Figure 4b. 

Because we had no prior knowledge as to whether MZA preferentially binds to the one terminus over the other in RSK1, we superimposed the two termini of RSK1. Figure 5c shows the overall NTKD and CTKD RSK1 crystal structures overlay with the amino acids in the ATP binding pocket of CTKD RSK1 labeled. It should be pointed out that there is notable similarity. Figure 5d shows a comparison of the active sites.

## 3. Discussion

The RSK family of proteins consists of four highly homologous isoforms (RSK1 to RSK4), with the exception of the N- and C-terminal sequences, where divergence is observed. RSKs can receive signals to their CTKD and in turn will transmit an activating signal to their NTKD. The two catalytic domains are connected via a conserved linker of approximately 100 amino acids. As noted earlier, NTKD belongs to the AGC family and is responsible for substrate phosphorylation. CTKD is homologous with the Ca^2+^/calmodulin-dependent kinase family and responsible for NTKD’s activation via autophosphorylation. 

Although the search for novel RSK inhibitors is ongoing, to our knowledge there is no RSK isozyme specific inhibitor. The RSK reversible dihydropteridinone BI-D1870 [24] and kaempferol glycoside SL0101 [25] inhibit the NTKD ATP-binding site with comparable potency. Similarly, the pyrrolopyrimidine FMK is a CTKD-irreversible inhibitor that binds to the ATP-binding pocket of RSK [26]. 

We have carried out an initial in vitro screening of the effect of MZA on 30 protein kinases. We have also compared the in vitro activity (inhibition activity range of 68% to less than 10%) with that of an in silico molecular docking experiment (binding affinity range of −11.8 to −6 Kcal/mol). The in vitro and in silico results indicate that MZA exhibits significant inhibitory activity toward RSK1 better than the previous study using commercially available inhibitors which were tested at 10-fold higher concentration [27,28]. 

We also wished to compare the in vitro results with those predicted by docking. However, multiple crystal structures exist. In order to identify representative macromolecular structures to employ in docking, we first superimposed the NTKDs of all crystal RSK1, CTKDs of RSK1, and repeated them with an overlay of the NTKDs and CTKDs of RSK2. Table 2 shows the structures selected for all subsequent docking experiments. Furthermore, even though the CTKD of RSK1 is an apo structure, its ATP-binding pocket has been described [33]. Figure 5a depicts the equivalent amino acids in the ATP-binding pockets of both RSK1 and RSK2 structures (CTKD of RSK1 PDB ID: 3RNY and CTKD of RSK2 PDB ID: 4D9T). Glu 496 is constrained in a specific orientation within the ATP-binding pocket of 3RNY, while Glu 500 of 4D9T is the corresponding residue (see Figure 5a and Table 3) [33]. A similar overlay was undertaken for the NTKDs of RSK1 and RSK2 (Figure 5b and Table 3). It can be seen that the ATP-binding pocket of RSK1 consists of Asp 142, Asp 205, Leu 144, and Asn 192, as reported by Ikuta et al. [34]. The active conformation of RSK2 is lined by polar charged residues (Asn 198, Asp 193, Lys 100, and Asp 211), along with a hydrophobic patch, including Lys 100, Phe 79, Leu 101, Lys 216, and Leu 214 [32]. The identified amino acids lining up respective binding pockets were employed in all subsequent docking experiments. Because MZA is shown herein to be selective toward RSK1 over RSK2, and since we had no prior knowledge whether it preferentially binds to the one terminus over the other, we superimposed the two termini of RSK1. Figure 5c shows the overall NTKD and CTKD RSK1 crystal structures overlay with the amino acids in the ATP binding pocket of CTKD RSK1 labeled. In comparing the two topologies, there is a notable similarity, in that, with the exception of the N-lobe consisting of a five-stranded antiparallel β sheet and a missing helix corresponding to αE of CTKD, the remaining structures superimpose rather well. Figure 5d shows a comparison of the active sites. All but one (Glu 454) of the CTKD amino acids correspond to similar type residues in NTKD. Specifically, Lys 447, Asp 557, Asp 535, and Asn 540 of CTKD are equivalent with Lys 94, Asp 205, Asp 187, and Asn 192 of NTKD. This appears to be in agreement with Fisher and Blenis, who provided evidence that both domains are active and that NTKD is responsible for phosphotransferase activity, whereas CTKD is catalytically active [35]. 

In comparing the in vitro and in silico results, it appears that MZA exhibits inhibitory activity toward RSK1, better than an earlier study which used commercially available inhibitors, tested at 10-fold higher concentration [27,28]. These biochemical observations were further validated in two in vitro cancer cell lines where MZA inhibited the protein expression of RSK1, RSK2, and phospho-RSK1. Although the cervical cancer cell lines SiHa and CaSki express total RSK2 protein, the anti-phospho-RSK2 antibody did not detect the phospho-RSK2 level and needed further validation. RSK3 and RSK4 were not evaluated in this study, as they are not part of the 30 protein kinases panel screened. 

A non-competitive allosteric binding-site may be present in the protein kinase, in addition to the competitive ATP-binding site. In the present study, MZA is docked to the competitive ATP-binding site only, since the location of the allosteric site is yet to be determined in many kinases. The predicted binding affinity of MZA is compared to ATP in order to decipher selectivity trends of MZA vs. ATP. Table 1 indicates that MZA is a potential inhibitor for a number of protein kinases, such as RSK1, LCK, GSK3β, NEK2α, SAPK2α, CSK, PHK, and SAPK3, when compared to ATP. However, despite the favorable binding affinity, upon closer inspection, MZA does not bind in the ATP-binding pocket in many of the protein kinases, as the binding pocket is narrow and the molecular structure of MZA is bulky. For example, MZA was predicted to bind to PDK1, even though it displayed poor in vitro activity. Upon visual inspection, it appears that MZA binds at the external opening of the binding pocket, thus occupying different binding space relative to ATP (Figure 6a). In contrast, the RSK1–MZA complex has MZA bound at the same locus of the binding pocket as ATP (Figure 6b). Predicted MZA-bound kinase complexes are shown in Appendix A. 

Our results suggest that MZA is a moderate, yet selective inhibitor of the RSK1 isozyme based on biochemical data and demonstrated a 10-fold selectivity in potency between RSK1 and RSK2 (relative IC_50_ values of 15.01 ± 1.94 μM and 108.40 ± 16.93 μM, respectively). In order to further explore MZA’s potential binding interactions and shed light on the observed selectivity toward RSK1 versus RSK2, a more tailored computational strategy was employed. 

Because we were not certain whether MZA binds to the N- or C-terminal domains, we opted to perform docking experiments toward both NTKD and CTKD of the two isoforms in order to (1) rationalize the observed selectivity between the two isozymes and (2) identify the plausible preferential binding for one domain over the other.

Docking of MZA into the CTKD of RSK1 revealed strong hydrogen bonding and π-stacking interactions (Figure 7a). Specifically, the hydroxyl hydrogen bonds with Asp 557 and Arg 454, while the nitrogen of aza-cyclooctene forms a hydrogen bond with Lys 537. Finally, aliphatic–aromatic interactions are observed between Phe 560 and the polycyclic moiety of MZA.

In contrast, the predicted binding mode of MZA with CTKD of RSK2 (Figure 7c) suggests fewer and weaker interactions with the binding pocket, which in turn could account for the observed higher affinity of the compound for RSK1. The hydroxyl group is oriented toward the backbone peptide bond nitrogen of Val 300, while the nitrogen interacts with the carboxylate of Glu 500. The remaining interactions are primarily hydrophobic in nature, and no aryl–alkyl interactions are predicted, since Phe 564 (equivalent to 560) is far from the pose. 

Similarly, we docked MZA into the NTKDs of both RSK1 and RSK2, as described in the preceding sections. We observed an orientation much deeper into the binding pocket in RSK1 and more interactions than in RSK2. Specifically, the hydroxyl group hydrogen-bonds with the backbone nitrogen of Gln 70, while the N of the aza-cyclooctene is in close proximity with Asp 148. Leucines 194 and 68 form a hydrophobic patch around the aza-carbazole, while aryl–alkyl interactions are observed with Phe 150 (Figure 7b). When MZA was docked into the NTKD of RSK2, limited interactions were observed. The hydroxyl interacts with Asp 154, and the secondary amine forms a hydrogen bond with the backbone of Leu 74. The π-π stacking interactions are observed between carbazole and Phe 149 (see Figure 7d). 

Consequently, it seems that MZA forms a tighter complex with RSK1 and is positioned deeper into the pocket contrary to RSK2. Thus, using docking experiments coupled with visual comparisons of reported crystal structures, we have explained observed binding affinities and selectivity of MZA toward RSK1. Another point of concern was whether we could predict the ligand’s binding to NTKD or CTKD. Because the MZA–NTKD RSK1 complex appears to be more stable due to stronger interactions with the surrounding amino acid residues in the ATP-binding pocket, in contrast to those observed in the MZA–CTKD RSK1 complex, we suggest that MZA preferentially binds to NTKD. Further, an examination of the predicted binding energies of the NTKD and CTKD complexes with MZA (models of −62.132 and −55.497, respectively) support the notion that MZA shows a preference toward the NTKD of RSK1.

In conclusion, we report herein an integrated biochemical, in vitro cellular, and computational study investigating the marine-derived alkaloid MZA. MZA was screened in a panel of 30 protein kinases (in vitro and in silico experiments). The data presented in Table 1, together with Appendix A depicting the predicted MZA–protein kinase complexes, provide an insight into the correlations between bioactivity and preferred binding configuration. With the addition of the kinases’ activity data to those previously reported, Manzamine is a potential multi-kinase inhibitor. Finally, we show that MZA selectively inhibits RSK1 over RSK2 and predicts that it binds to its N-terminal kinase domain. To our knowledge, this is the first report of RSK1 as a new macromolecular target of MZA, a finding that extends the pharmacology of the manzamine-type alkaloids. The in vitro results combined with the predicted RSK1–MZA complexes provide the foundation for future rational design strategies employing in silico modeling, synthesis, and structure–activity relationships. In turn, a better understanding of MZA’s pharmacology may be attained, and it may find utility in chemical biology studies involving RSK1 expression. Manzamine A’s interaction with a number of targets makes it difficult to predict in whole-cell models if cellular peturbations after MZA treatment are due to RSK1 vs. other targets. However, the data presented reveal the utilty of MZA in the design of future RSK1 selective inhibitors. 

## 4. Materials and Methods

### 4.1. Materials

Manzamine A (MZA) (Figure 1) was isolated as part of the SeaPharm project from a marine sponge species of the genus Haliclona collected off Manzamo, Okinawa, Japan, as described [9]. MZA was provided by Dr. Amy Wright, Florida Atlantic University Harbor Branch Oceanographic Institute, Fort Pierce, Florida, for this study. The NMR spectra of MZA that are included in the Appendix A demonstrate that MZA was > than 98% pure. The reversible dihydropteridinone BI-D1870, an NTKD-interacting ATP-binding site-specific RSK inhibitor was provided by P. Cohen, University of Dundee, Scotland, UK [26]. A 10 mM stock of BI-D1870 and MZA were prepared in DMSO and stored at −80 °C prior to use in the experiments.

### 4.2. Protein Kinase Activity Assays

Thirty protein kinase activity assays were performed by the Kinase Profiling Screening Service, University of Dundee, as described elsewhere [30]. MZA (1 μM) was used in all protein kinase assays. Results are presented as a percent inhibition of kinase activity of control incubations (average of duplicate determinations) and are tabulated in Table 1. 

### 4.3. RSK1 and RSK2 Protein Kinase Assays 

Rat RSK1 (Genbank M99169) and human RSK2 (Genbank NM_004586) protein kinase assays were performed as described in Reference [26], but with the following modifications: RSK1 and RSK2 (10 mU) were diluted (20 mM MOPS pH 7.5, 1 mM EDTA, 0.01% Brij35, 5% glycerol, 0.1% β-mercaptoethanol, 1 mg/mL BSA) and assayed against KKLNRTLSVA in a final volume of 25 µL, containing 50 mM Na-β-glycerophosphate, pH 7.5; 0.5 mM EDTA; 30 µM substrate peptide; 10 mM magnesium acetate; 0.05 mM (^33^P-g-ATP) (50–1000 cpm/pmole); and 5 μL MZA, BI-D1870 or vehicle, and then incubated for 40 min at room temperature. The kinase assays were stopped by adding 5 µL of 0.5 M (3%) orthophosphoric acid, and 10 μL of the sample was transferred to labeled P81 filter-paper circles and allowed to bind for 30 s. Circles were then washed for five minutes with a wash buffer (50 mM orthophosphoric acid), transferred to scintillation vials containing 5 mL of scintillation cocktail, and read in a scintillation counter.

### 4.4. Computational Methods 

#### 4.4.1. Molecular Docking with AutoDock Vina

The MM2 energy-minimized 3D structures of ligands of MZA and ATP were optimized by using ChemBio3D Ultra version 12.0 (PerkinElmer Informatics, Waltham, MA, USA). The crystal structure of protein kinase was obtained from the Protein Data Bank (www.rcsb.org, accessed on 24 December 2020) [34,35]. The ligands and receptor for molecular docking experiments were prepared by using AutoDockTools version 1.5.6 (The Scripps Research Institute, La Jolla, CA, USA), in which the polar hydrogens were added to these structures [41,42]. The grid box parameter was set to cover the binding pocket in the protein kinases (Table 4). Docking was performed by using AutoDock Vina, and the outputs were visualized and analyzed with BIOVIA Discovery Studio Visualizer version 17.2.0 (Dassault Systèmes, Vélizy-Villacoublay, France). Predicted ligand–receptor binding affinities are tabulated in Table 1.

#### 4.4.2. Selection of Targets and Identification of Critical Amino Acids

The atomic coordinates of Ribosomal S6 Kinase 1 (RSK1) and Ribosomal S6 Kinase 2 (RSK2) were obtained from the Protein Data Bank (PDB). Specifically, complexes with staurosporine (PDB ID: 2Z7R) [32], purvalnol A (PDB ID: 2Z7S) [32], and AMP-PCP (PDB ID: 2Z7Q) [32] were considered for the N-terminus of RSK1. Similarly, for the N-terminus of RSK2, complexes with afzelin (PDB ID: 4EL9) [39], the flavonoid glycoside quercitrin (PDB ID: 4GUE) [40], flavonoid glycoside SL0101 (PDB ID: 3UBD) [39], and an active conformation of RSK2 (PDB ID: 3G51) [31]. In regards to the C-terminus of RSK1 the apo structure (PDB ID: 3RNY) [37] was employed. Finally, representatives of the C-terminal RSK2 were an apo structure (PDB ID: 2QR8) [43], and complexes with (E)-methyl 3-(4-amino-7-(3-hydroxypropyl)-5-p-tolyl-7H-pyrrolo[2,3-d]pyrimidin-6-yl)-2-cyanoacrylate (PDB ID: 4D9T) [38] and (E)-tert-butyl-3-(4-amino-7-(3-hydroxypropyl)-5-p-tolyl-7H-pyrrolo[2,3-d]pyrimidin-6-yl)-2-cyano- acrylate (PDB ID: 4D9U) [38].

Structural comparisons were carried out within the Maestro interface (Schrodinger, LLC, New York, NY, USA) in order to (1) select the structure that would be similar to the majority of representatives of each terminus (Table 2) and (2) identify critical residues that would later serve to define respective binding pockets. Amino acids lining the binding pockets of the crystal RSK1 and RSK2 structures reveal a high degree of similarity (Table 3).

#### 4.4.3. Protein Preparation

The receptors were prepared by assigning bond orders, adding hydrogens and finding overlaps, followed by hydrogen bond optimization and minimization, using the Protein Preparation Wizard. Minimization employed the ‘Impref’ utility, which runs a series of constrained impact minimizations with gradually decreasing strength of the heavy-atom restraining potential. Two minimizations were initially performed with the heavy-atom restraint potential force constant at 10. In the first minimization, the torsional potential was turned off to improve hydrogen optimization, whereas the second minimization restored the torsional potential. The restraining potential force constant was subsequently reduced to 3, 1, 0.3, and 0.1. If the output structure from a minimization exceeds the specified RMSD threshold, relative to the starting structure, the program stops and returns the structure from the previous minimization. Thus, the RMSD is checked at the end of each round of minimization. Receptor grid generation was subsequently employed with the van der Waals radius scaling factor set to 1.0, and the partial charge cutoff was set at 0.25.

#### 4.4.4. Docking Glide 5.7

Glide version 5.7 (Schrodinger, New York, NY, USA) [33,36] was used in this study. Glide grids were generated with ligand scaling of 0.8 for the van der Waals radii. Ligands were docked and ranked by using standard precision, followed by post-docking minimization. All docking experiments were performed with default settings, and 30 poses per ligand were saved for further consideration. Poses were visually inspected and analyzed based on the native scoring function and binding interactions.

### 4.5. Immunoblotting for RSK1 and RSK2 in SiHa and CaSki Human Cervical Carcinoma Cancer Cell Lines

SiHa and CaSki human cervical carcinoma cell lines were obtained from ATCC and maintained as described previously [18]. SiHa (HTB-35) and CaSki (CRL-1550) cells were plated at a seeding density of 1 × 10^6^ cells in a 100 mm tissue culture Petri dish and allowed to adhere overnight. The next day, the cells were treated with 2 and 4 µM concentrations of MZA and incubated for 48 h. We also tested a higher concentration (4 and 6 µM) of MZA on the effect of phospho-RSK1 and phospho-RSK2 after 48 h of treatment. Following incubation, the cells were harvested by trypsinization and washed with 1×PBS, and then protein was extracted in radioimmunoprecipitation assay (RIPA) buffer (Sigma; 20-2188, St. Louis, MO, USA) containing a cocktail of protease inhibitor (Sigma; P8340). Following protein quantification, an equal amount of protein was loaded on to SDS–polyacrylamide gel and transferred electrophoretically to PVDF membrane. The membrane was blocked in 5% BSA in PBS containing 0.01% Tween-20 and incubated with primary antibody against RSK 1 (Santa Cruz, Dallas, TX, USA; sc-393147) or RSK2 (Santa Cruz; Dallas, TX, USA; sc-9986) at a dilution of 1:250, phospho-RSK1, and phospho-RSK2 (Santa Cruz, Dallas, TX, USA; sc-377526 and sc-374664) at a dilution of 1:50 or GAPDH (Cell signaling, Danvers, MA, USA) at a 1:1000 dilution. The membrane was washed in TST and incubated with secondary IgG HRP-conjugate at 1:2000 dilution. Protein expressions were visualized by using chemiluminescence kit (BioRad, Hercules, CA, USA) on an invitrogen iBright FL1000 imaging system, and the expected protein band size was compared with a standard protein marker. The densitometry analysis was performed on scanned immunoblot images, using the Image J software analysis tool (IJ 1.52a, National Institute of Health, USA).

### 4.6. Statistical Analysis

Data were statistically analyzed with Prism^®^ software package version 9 from GraphPad, San Diego, CA, USA. For Figure 2, data were expressed as mean ± SEM from 3 independent experiments (*n*), each experiment with triplicate determinations. Data were analyzed with Prism^®^ software package version 9 from GraphPad, San Diego, CA. One-way analysis of variance (ANOVA), followed by Dunnett’s post hoc procedure, was performed on all sets of data. In addition, statistical significance between the dose-dependent effect of MZA on RSK1 vs. RSK2 was determined by using two-way ANOVA. Differences were considered statistically significant at *p* < 0.05. For Figure 3, to examine the dose-effect of MZA on RSK1 and RSK2 protein expression levels, ANOVA with Tukey’s multiple comparisons test was used and reported *p*-value comparisons between the control and high MZA dose.

## Figures and Tables

**Figure 1 marinedrugs-19-00506-f001:**
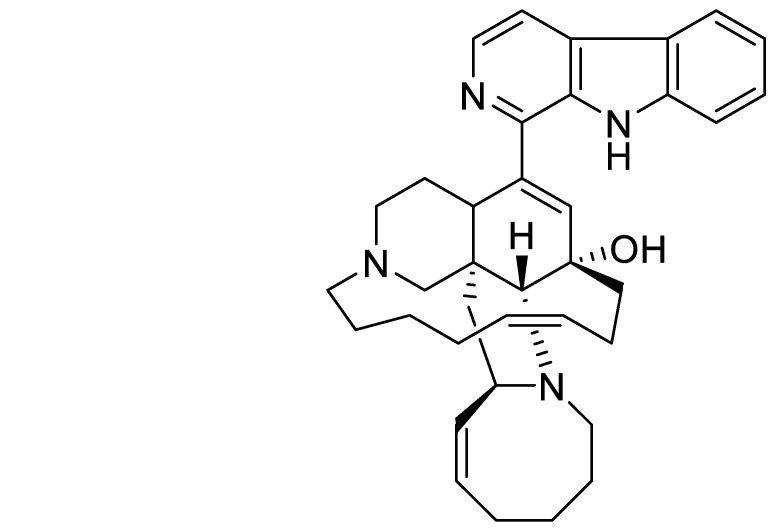
MZA is an indole-derived alkaloid isolated from the marine sponge *Haliclona* sp. [4].

**Figure 2 marinedrugs-19-00506-f002:**
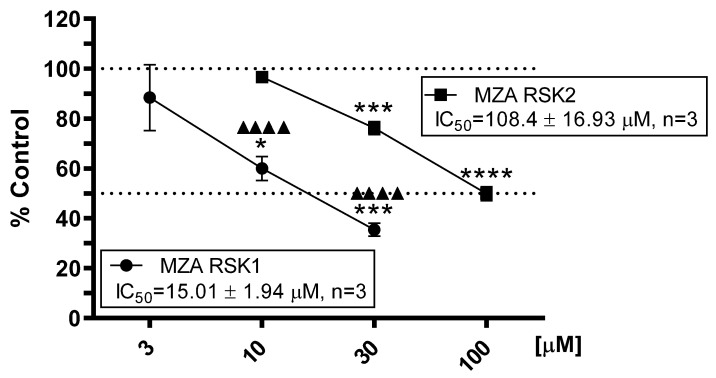
Differential effect of MZA on RSK1 and RSK2 activity. RSK1 and RSK2 kinase assays were conducted as described in Materials and Methods. Data expressed as mean ± SEM of three independent experiments (*n*), each experiment with triplicate determinations; * *p* < 0.05, *** *p* < 0.01, and **** *p* < 0.001 vs. control (DMSO). ^▲▲▲▲^ *p <* 0.001 RSK1 vs. RSK2.

**Figure 3 marinedrugs-19-00506-f003:**
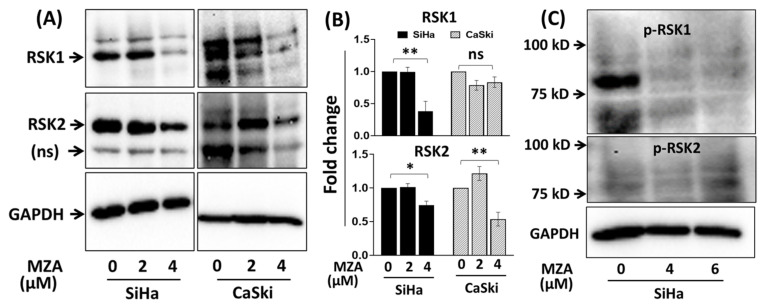
SiHa and CaSki human cervical carcinoma cells were treated with 2 and 4 µM of MZA for 48 h and protein expression of RSK1 and RSK2 was determined by Western blot assay. (**A**) Representative blots of RSK1 and RSK2 expression, (**B**) histograms from the pooled data for RSK1 and RSK2 repeats (*n* = 3), and (**C**) the use of higher concentration of MZA (4 and 6 µM) for 48 h and its effect on phospho-RSK inhibition. Data represented as mean ± SE of three independent experiments; * *p* < 0.05 and ** *p* < 0.01; ns = non-specific.

**Figure 4 marinedrugs-19-00506-f004:**
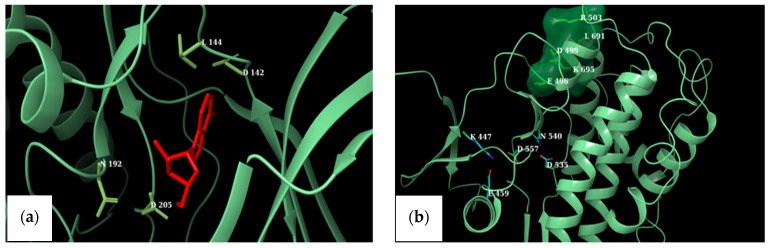
(**a**) ATP-binding site in the NTKD of RSK1 with the AMP-PCP ligand shown in red. (**b**) CTKD of RSK1 with ATP-binding site residues depicted by atom-type, while green residues reflect amino acids that constrain Glu 496 in a favorable conformation pointing toward the ATP-binding pocket.

**Figure 5 marinedrugs-19-00506-f005:**
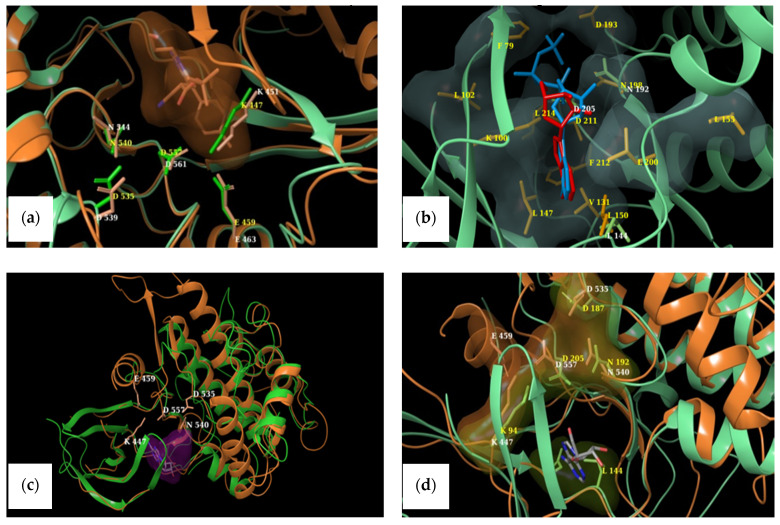
(**a**) Corresponding amino acids in the ATP-binding pockets of the C-termini of RSK1 (green) and RSK2 (orange). Residues in RSK1 are labeled yellow, while white labels designate RSK2 residues. (**b**) The NTKDs of RSK1 (green, white labels) and RSK2 (orange, yellow labels). Only the ribbons of RSK1 are displayed. Bound ligand in the NTKD RSK1 is depicted in red, while the surface of the binding pocket of RSK2 is shown with its respective bound ligand in cyan. (**c**) Overall topologies of the NTKD (green ribbons) and CTKD (orange, white labels) of RSK1 superimposed. The surface of the ligand in CTKD is in purple. (**d**) Close-up view of the binding pocket. Amino acids in NTKD are labeled in yellow. All but one residue (Glu 459) thought to be catalytic in CTKD have equivalent ones in NTKD. Surfaces of residues are also depicted in orange (CTKD) and green (NTKD).

**Figure 6 marinedrugs-19-00506-f006:**
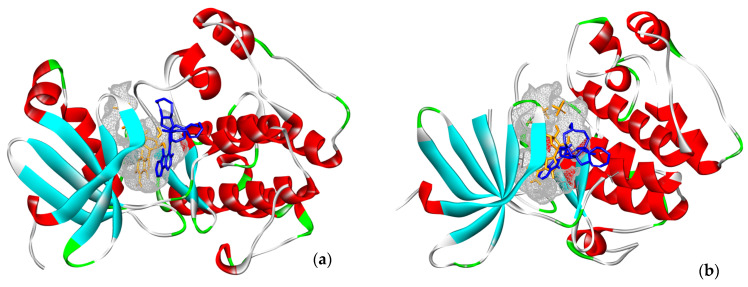
MZA (blue) and ATP (orange) is shown docked to (**a**) PDK1 (PDB ID: 5LVL) and (**b**) RSK-1 (PDB ID: 2Z7Q) protein kinases (binding pocket is shown as gray mesh).

**Figure 7 marinedrugs-19-00506-f007:**
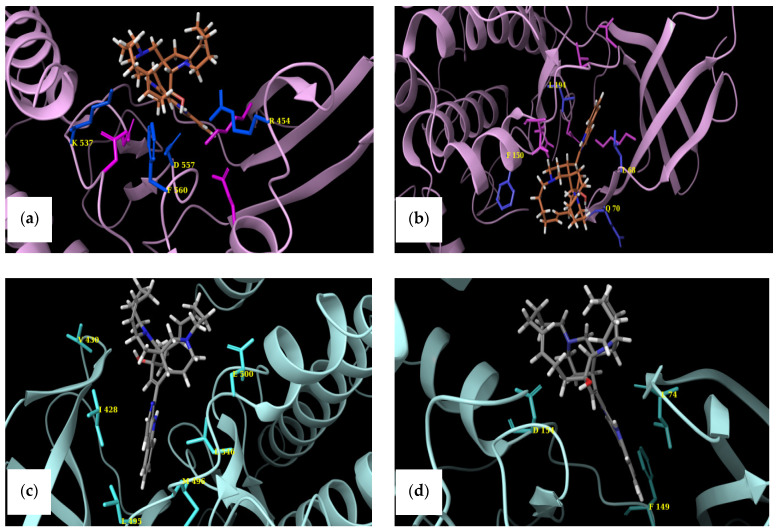
Docked pose of MZA into the (**a**) CTKD of RSK1 and (**b**) NTKD of RSK1. Amino acids forming binding interactions with the ligand are shown in blue and labeled. Catalytic residues reported in the literature are shown in dark purple sticks for reference in regards to MZA’s positioning within the binding pocket. Predicted pose of MZA into the (**c**) CTKD of RSK2 and (**d**) NTKD of RSK2. Amino acids forming bonding interactions are depicted in sticks and labeled.

**Table 1 marinedrugs-19-00506-t001:** Effect of MZA on a 30 protein kinases enzyme panel and in silico binding affinity.

No.	Protein Kinase	Full Name	MZA 1 μM *	PDB ID	Predicted Binding Affinity (Kcal/mol)	
					ATP	MZA
**1**	**MAPKAP-K1α (RSK-1)**	**MAPK-Activated Protein Kinase-1α**	**68**	**2Z7Q**	**−7.6**	**−9.5**
2	SGK	serum and glucocorticoid-induced kinase	39	2R5T	−9.3	−8.6
3	LCK	lymphocyte kinase	29	3MPM	−8.6	−10.9
4	MSK1	mitogen and stress-activated protein kinase-1	27	3KN5	−8.3	−8.5
5	GSK3β	glycogen synthase kinase-3β	27	1Q5K	−7.4	−8.5
6	NEK 6	NIMA-related protein kinase 6	26	n/a		
7	SAPK2α/P38	stress-activated protein kinase-2α	25	5OMH	−7.1	−9.1
8	PKBα	protein kinase B	25	3CQW	-8.9	−6.9
9	JNK1a1	c-Jun N-terminal kinase	20	1UKI	−7.6	−7.9
10	S6K1	p70 ribosomal protein S6 kinase	18	3A60	−7.9	−8.6
11	DYRK1A	dual tyrosine phosphorylated and regulated kinase 1A	18	2VX3	−8.2	−7.9
12	CK2	casein kinase-2	17	1J91	−7	−8
13	SAPK2β/p38β2	stress-activated protein kinase-2β	16	n/a		
14	CHK1	checkpoint kinase-1	14	3U9N	−7.9	−8.2
15	CSK	C-terminal Src Kinase	11	1BYG	−7.7	−9.2
16	MKK1	MAPK kinase (mitogen-activated protein kinase)	10	1S9J	−8.9	−9.4
17	PKA	cyclic AMP-dependent protein kinase	10	2JDS	−9.1	−7.9
18	PKα	protein kinase C	10	3IW4	−8.5	−8.7
19	CK1	casein kinase-1	10>	1CSN	−8.3	−8.9
20	MAPKAP-K2 (RSK-2)	MAPK-activated protein kinase-2	10>	3G51	−9.5	−7.6
21	MAPK2/ERK2	mitogen-activated protein kinase	10>	2OJJ	−8.6	−7.9
22	PRAK	p38-regulated/activated kinase	10>	n/a		
23	NEK2α	NIMA-related protein kinase 2α	10>	5M53	−7.8	−9.9
24	AMPK	AMP-activated protein kinase	10>	2YA3	−9	−9.6
25	SAPK4/P38d	stress-activated protein kinase-4	10>	5EKN	−7.8	−7.4
26	PHK	phosphorylase kinase	10>	2Y7J	−7.1	−9.6
27	SAPK3/P38γ	stress-activated protein kinase-3	10>	1CM8	−7.5	−11.8
28	PDK1	3-phosphoinositide-dependent protein kinase-1	10>	5LVL	−8.9	−8.8
29	ROCK-II	Rho-dependent protein kinase	10>	6ED6	−8.5	−6
30	CDK2/cyclin A	cyclin-dependent kinase 2-cyclin A complex	10>	4EZ7	−8.8	−7.2

* Mean % inhibition; n/a—not available.

**Table 2 marinedrugs-19-00506-t002:** Crystal structures employed in the study.

RSK1		RSK2	
*CTKD*	*NTKD*	*CTKD*	*NTKD*
3RNY	2Z7Q	4D9T	3G51

**Table 3 marinedrugs-19-00506-t003:** Corresponding amino Acids in the binding pockets of NTKDs and CTKDs of RSK1 and RSK2.

*RSK1* [32]	*RSK2* [31,36]
*N-Terminal*	
Phe 73 ^e^	Phe 79
Asp 142 ^a^	Asp 148 ^b^
Leu 144 ^c^	Leu 150 ^c^
Lys 94 ^c^	Lys 100 ^c^
Asp 205 ^c^	Asp 211 ^c^
Asn 192 ^c^	Asn 198 ^c^
Val 125 ^e^	Val 131
Asp 187 ^d^	Asp 193
***RSK1* [37]**	***RSK2* [38]**
*C-Terminal*	
*Residues around Glu496*	*Equivalent residues*
Leu 691	Pro 696
Glu 496	Glu 500
Arg 503	Arg 507
Asp 499	Asp 503
Cys 432	Cys 436
Cys 556	Cys 560
Thr 489	Thr 493
*ATP-binding pocket*	*Equivalent residues*
Lys 447	Lys 451
Glu 459	Glu 463
Asp 557	Asp 561
Asp 535	Asp 539
Asn 540	Asn 544

^a^ From Reference [39]. ^b^ Residues in RSK2 that are equivalent to residues reported in Reference [39] for RSK1. ^c^ Equivalent residues reported in References [39,40]. ^d^ Residue in RSK1 corresponding to the one reported for RSK2 [40]. ^e^ Reference in RSK1 that is equivalent to the one reported for RSK2 [34].

**Table 4 marinedrugs-19-00506-t004:** Protein kinases docking grid box parameter ^a^.

PDB ID	X-Center	Y-Center	Z-Center
1BYG	27.128	45.519	13.045
1CM8	41.771	75.375	4.443
1CSN	−24.576	49.359	51.847
1J91	−11.675	3.876	3.645
1Q5K	22.619	22.938	8.581
1S9J	32.089	35.875	41.601
1UKI	1.562	39.364	29.179
2JDS	9.808	9.157	−0.179
2OJJ	−13.28	13.521	38.972
2R5T	33.954	32.472	64.595
2VX3	39.08	22.423	−37.766
2Y7J	−14.917	−6.949	40.443
2YA3	8.821	10.086	30.369
2Z7Q	0.188	−3.734	23.374
3A60	−6.672	3.557	38.526
3CQW	4.789	2.807	17.128
3G51	23.207	32.364	87.888
3IW4	4.829	30.195	53.056
3KN5	25.772	36.512	76.29
3MPM	25.455	11.311	51.812
3U9N	12.605	−3.555	10.933
4EZ7	−1.982	8.889	−28.626
5EKN	−12.503	12.799	−28.136
5LVL	−14.083	−4.449	−8.027
5M53	31.849	−4.082	13.028
5OMH	−2.482	−1.883	−19.355
6ED6	26.752	45.831	53.583

^a^ The grid box spacing was set as 1.0 Å with x-dimension = 20; y-dimension = 20; and z-dimension = 20.

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
