# Peer review of "RSK1 vs. RSK2 Inhibitory Activity of the Marine β-Carboline Alkaloid Manzamine A: A Biochemical, Cervical Cancer Protein Expression, and Computational Study"

_marinedrugs, 2021, doi:10.3390/md19090506_

Round 1

Reviewer 1 Report

The manuscript by Mayer et al describes the discovery of the inhibitory activity of MZA on RSK1. The authors demonstrated this enzymatically and exploited docking study to predict the potential binding mode and selectivity of the inhibitor. In addition, the cellular activities of MZA in two cell lines were tested, but not conclusive. Although the manuscript initially set out to provide an interesting finding, throughout there is little convincing data, hence little scientific evidence to support the claim. There are many open questions in nearly all aspects, which thus weaken the authors’ conclusion of the potential inhibition of the kinase by MZA. This includes for example the followings:

Screening of the inhibitor at 1 μM shows 68% inhibition of RSK1. However, the ‘dose-response’ experiment suggests IC50 of 10 μM. If at 1 μM could reduce nearly half of the kinase activity, how could IC50 turn out to be 10-fold higher? In fact, the calculated IC50s from just only three points with a drawn straight line are highly promiscuous (and even for RSK2 the predicted IC50 is outside the range of the used MZA concentrations). A full dose-response experiment is needed for a more accurate IC50. In addition, the exploited enzymatic assay seems to be for the N-terminal kinase domain of RSK1 (as the authors referred to the previous reference as well as described in the method for testing the inhibition of BI-D1870 in parallel, but no data available?). If so, is this not indicating an inhibition of the N-terminal kinase domain? None of these is explained. Furthermore, there is little evidence to support a conclusion of the N-terminal kinase domain of RSK1 as the main target. Further characterisation is needed for this.

The authors then tested MZA in some cell lines, and probed RSK1 and RSK2 protein levels, leading to the claimed that the reduced protein levels are likely due to the effect of MZA on targeting RSK1/2. This is extremely invalid. To probe the effect of the kinase inhibition, it should be better to probe phosphorylation of the downstream targets or using other assays to detect on-target inhibition (e.g. NanoBret). The performed cellular experiments in this manuscript do not provide such evident that RSK1/2 are targeted by MZA in cells, rather the overall promiscuous/broad effects of MZA which is known to have diverse cellular activity (potential cytotoxicity?). In addition, 4 μM concentration is much weaker than enzymatic IC50 of 10 μM. One would expect much weaker cellular activity of an inhibitor.

Further study was to predict the binding mode in RSK1 and to provide rationale for the selectivity over RSK2. This constitutes the major part of the manuscript, and is disappointingly extremely vague, inconclusive and confusing with most sentences/parts also very repetitive throughout. It is extremely surprising that no details of predicted binding mode provided, rather vague discussion (which in fact should be the results) in a repetitive loop that does not point to any potential evidence for the interaction or the observed compound selectivity. In addition, there are many beta-carboline-bound kinase structures as well as the structures of GSK3b and CDK5 known as a target of MZA, and none of these were used for comparison. The author attempted to compare the details of the binding pockets between RSK1 and 2 and between their N- and C-terminal kinase domains, leading to some residues pointed out. From the kinase point of view, this part especially is very vague. Most residues in questions are a set of the highly conserved residues that are typically shared among most kinases and are essential for kinase activity, therefore they would unlikely to provide selecitivity. In addition, some other residues do not contribute significantly to the feature of the binding pocket (e.g. their side chains are outside), thus they again are unlikely to form a key reason. Furthermore, discussion on certain features such as beta strand 1 and 2 or αE is completely misleading. For example, the missing beta1 and 2 in RSK2 is not because the kinase does not have it, rather it is the consequence of disordering due to the presence/absence/effect of a ligand in the binding pocket. The additional helix in C-terminal kinase domain is not αE. Altogether, as docking could also be subjective there is no valid evidence to support the claim of the binding of MZA in RSK1 N-terminal kinase domain let alone its selectivity over RSK2 or other kinases.

Reviewer 2 Report

Solid paper describing new targets of manzamines, well known marine sponge-derived polycyclic β-carboline alkaloids with broad biological activities including inhibition of a 90 kDa ribosomal protein kinase S6 (RSK1), a vertebrate family of cytosolic serine-threonine kinases. The work included screening against a panel of 30 protein kinases, for which manzamine was found to have a moderate effect (68% inhibition) on RSK-1, versus little effect (<30% inhibition) on most other targets investigated. In vitro RSK kinase assays demonstrated a 10-fold selectivity in the potency of MZA against RSK1 versus RSK2, albeit the potency is in the moderate level, micromolar range (15-108 uM range).  

The effect of manzamine on inhibiting cellular RSK1 and RSK2 protein expression was validated in SiHa and CaSki human cervical carcinoma cell lines, showing strong reduction of protein expression after 48 hours at 4 uM of compound.  

The work was further supported by computational docking experiments highlighting stronger interactions between RSK1-manazamine than with RSK2, with stronger binding to the N-terminal kinase domain of RSK1 over the C-terminal domain.

Large volume of work included demonstrating new potential enzyme target of this well-known class of marine natural products though noting the potency in both the computational study and in vitro assays are in the moderate range though show good isoform selectivity for RSK-1. It is good to see the authors using the appropriate term “moderate” throughout the paper, as too many other papers try to inflate in words what the data says. This paper is more direct about the data and what it says, which should be commended and is certainly a worthy addition to this journal.      

Queries:

  • What is the effects of manzamines on other RSK isoforms (RSK-3/4), please include in discussion?
  • Formatting – Table 1 should appear on one page.
  • Table 1 column – show as % inhibition not % activity remaining, comes across as trying to make the results better than they are. In the text 68% inhibition is referred to but in the Table 32% activity remaining. Please correct the way the Table data is presented. In the Table column heading 1. M* (what does 1 refer to? 1uM?). Add to caption that testing was done at 1 uM.
  • In the discussion (around line 177) include the range of activities observed for both the calculation of activity and binding affinities (e.g. binding affinities in the range -6.9 to -10.9 kcal/mol), a comparison to gold standards in the field and then an indication of which proportion of your results would then be considered active/promising.   
  • Table 1- Include errors / STD related to both the calculation of activity and binding affinities.
  • 3 – what is the effect at shorter time periods such as 24 hours and were higher amounts than 4 uM tested – this seems like a very narrow range to be testing over?
  • Combine Figs 5 and 6 into one 4 panel figure.
  • Combine Figs 8 and 9 into one 4 panel figure.
  • Line 184 – manzamine is a potential inhibitor for a number of protein kinases – please indicate which ones and why (other than comparison to ATP). Overall these binding affinities are not strong compared to drugs on the market. Please support your statements here with clinically used examples and refs.
  • Line 181 – please expand on this statement. It is common that computational results are stronger than in vitro (and are used more to guide what assays are then performed) but at the end of the day, compounds need to be active in enzyme assays and ultimately cell and animal studies to proceed to the clinic. Expand further on why the in vitro results are not as strong, what modifications to the stucture could be made, and where would this work head in the future?
  • Explain how the in vitro assays can be fed back into the computational models to strengthen them and more accurately reflect future in vitro work?
  • Materials – how was the compound characterised and did it match reported data? Please include a copy of HPLC trace and/or NMR data showing purity of sample used and characterisation.
  • Table 3 caption grid box (not grit)
  • Refs need to be checked for consistent and journal appropriate formatting. Some refs have DOI’s others not?

Reviewer 3 Report

     The work is a carefully made, graphically attractive, well written and interesting, with potential biomedical applications.  It shows scientific soundness and appears with a good quality of the presentation.  Among other, it shows a novel and specific binding and inhibition of Manzamine A (MZA) to the SK1 protein kinase, predicting is done at the N-terminal kinase domain, unveiling a new macromolecular target for MZA and extending the pharmacological opportunities for this marine alkaloid.

     This reviewer would recommend revising the followed experimental procedures in order to describe them with more detail than is now in the manuscript, in which such details are just referred to previous publications.  Thus, in lines 293 and 300 of Materials and Methods (sections 4.1, and 4.2), dealing with Manzamine A isolation and with protein kinase activity assays, it is said that:   "... as described...", and "... as described elsewhere ...", in stead of minimally explaining the procedure, making the consultation to other references obligatory for the reader in order to minimally understand the work. This is not adequate, because the paper has to be understandable by itself.

    Also, it would be useful to better indicate the reasons to select the collection of 30 protein kinases assayed, besides the RSK ones

Round 2

Reviewer 1 Report

In the revised version of the manuscript, most part remains inconclusive. The major point is the claim of MZA as a RSK1 inhibitor that still does not have a strong evidence to support it. There is no doubt that the screening using a panel from Dundee's facility suggests inhibition of RSK1. However, there is no rigid follow-up experiment to really proof this. Several large kinase screenings often show false positive/negative hits (even well established facilities from diverse commercial sources), thus rigid follow-up experiment is needed, and this remains a major gap in this study. Therefore, the claim of MZA as a 'selective' RSK1 inhibitor presenting a tool for chemical biology as said in the abstract is rather vague.

The authors heavily relied on the results from the in silico study. In fact, there are many kinases that showed good MZA binding such as LCK, yet poor biochemical inhibition. Therefore, without solid experimental data it is unconvincing that MZA could provide a tool for biological study of RSK1. In addition, the cellular inhibition remains inconclusive. The authors still have not ruled out potential cytoxicity of the compound. The used concentration at 4 uM remains much lower than 'predicted' IC50, which does not suggest that RSK1 is the main target in this cellular setting. It is very unlikely that cellular IC50 can be much lowe than enzymatic IC50. Overall, this therefore does not suggest/support the claim that the effects in cellular model seen here is due to RSK1 inhibiton by MZA.

The prediction/docking study remains elusive. Most residues mentioned are conserved residues, which are thus unlikely to provide a basis for selectivity. This section therefore provide little information and remains largely unclear. The focus of the docking study is to probe potential binding mode of MZA, thus the authors should focus on presenting closed-up details of the binding pocket with clear molecular interaction details of the contribution of 'claimed' residues. In fact, as previously most of the mentioned residues do not have their side chains contributing to the binding pocket, thus it is unclear how these residues can act as selectivity filters. In addition, MZA shares a core scaffold with harmine which is a known DYRK inhibitor and has its binding mode in the kinase determined. This previous information could be useful for prediction of MZA's binding mode. In addition, it has been shown previously that this class of natural product can exert activity toward other protein such as MOA, presenting undesirable cytotoxicity, thus the cellular effects observed for MZA remain questionable. In addition, in general most of this section which described the results from docking study should be included in the results rather than discussion.

Author Response

Manuscript MARINE DRUGS # 1305520: Response to Reviewer # 1

RSK1 vs RSK2 inhibitory activity of the marine β-carboline alkaloid Manzamine A: a biochemical, cervical cancer protein expression and computational study

Alejandro M.S. Mayer, Mary L. Hall, Joseph Lach, Jonathan Clifford, Kevin Chandrasena, Caitlin Canton, Maria Kontoyianni, Yeun-Mun Choo, Dev Karan, and Mark T. Hamann.

We thank reviewer 1 for the additional review of our manuscript and comments. We have gone through our revised manuscript very carefully once again and wish to provide the following response to reviewer 1’s comments.

We do not agree with reviewer 1's assessment that our data does not strongly support that manzamine A is a selective inhibitor of RSK1 for the following reasons: 1) Table 1 shows that at 1 µm concentration manzamine is a 68% inhibitor of RSK1 vs 10% for RSK2;  2)  Table 1 additionally  provides the results of Manzamine A docking vs ATP with a -9.5 for RSK1 vs -7.6 Kcal/mol for RSK2 and ATP;  3) Figure 2  shows a relative IC50=15.01±1.94 μM for the purified RSK1 enzyme vs relative IC50=108.40±16.93 μM for the purified RSK2 enzyme for manzamine A, consolidating data from three independent experiments with triplicate determinations and various concentrations;  4) The data presented in Table 1 and Figure 2 are then further supported in Figure 3 using SiHa and CaSki human carcinoma cells where protein expression of RSK1 and RSK2 was inhibited by Manzamine A. With regards to reviewer 1’s comment that “the authors… used concentration at 4 uM remains much lower than 'predicted' IC50, which does not suggest that RSK1 is the main target in this cellular setting. It is very unlikely that cellular IC50 can be much lower than enzymatic IC50”, we agree with reviewer 1 that although the inhibitory concentration of Manzamine A on RSK1 and RSK2 protein expression in SiHa and CaSki human carcinoma cells was lower than the IC50 with purified RSK1 enzyme, the published literature reports similar observations, e.g. with the NSAID ibuprofen, Mitchel et al. reported that ”…Ibuprofen was more potent as an inhibitor of COX-2 in intact cells than in either broken cells or purified enzymes” (underline is ours), Mitchell et al.   Selectivity of nonsteroidal antiinflammatory drugs as inhibitors of constitutive and inducible cyclooxygenase, Proc. Natl. Acad. Sci. U S A. 1993 Dec 15;90(24):11693-7.  doi: 10.1073/pnas.90.24.11693.  Finally, with regards to reviewer 1’s comment that “The authors still have not ruled out potential cytotoxicity of the compound…”, as explained in the earlier response to reviewer 1 and published in Ref#16, the predicted IC50 of MZA was analyzed over a broad range of concentration at different time points, and the calculated IC50 of 4uM was the best selected dose of Manzamine A to perform SiHa and CaSki human carcinoma cells protein expression of RSK1 and RSK2. If we had used a higher concentration of MZA, MZA becomes toxic to the cells. Thus we are unclear to what potential toxicity the reviewer 1 is referring to.

In our opinion our revised manuscript is the most comprehensive and thorough set of experimental studies at the biochemical, cellular and computational levels to validate RSK1 vs RSK2 kinase selectivity for Manzamine A.  Reviewer 1 provides a very vague assessment that somehow the data is not adequate leaving the impression that he/she did not read our manuscript carefully, and furthermore, there are no specific experiments suggested by reviewer 1.  We certainly agree that the cellular reduction in RSK1 activity and expression could indeed be due to another interaction, however when the revised manuscript data are supported by two biochemical studies, a cellular protein expression study, and computational modeling studies from two labs, there is little doubt that selective RSK1 inhibition by Manzamine A is evident. We do not believe we can do a better job of validating the RSK1 vs RSK2 inhibitory activity of the marine β-carboline alkaloid Manzamine A than what we already have extensively revised and submitted.  We are hopeful or manuscript will be acceptable for publication in Marine Drugs. However, should our manuscript not be accepted for publication in Marine Drugs, we will understand your decision, and will submit this revised manuscript to another journal.